# Involvement of Cytokines in the Pathogenesis of Diabetic Macular Edema

**DOI:** 10.3390/ijms22073427

**Published:** 2021-03-26

**Authors:** Hidetaka Noma, Kanako Yasuda, Masahiko Shimura

**Affiliations:** Hachioji Medical Center, Department of Ophthalmology, Tokyo Medical University, 1163, Tatemachi, Hachioji, Tokyo 193-0998, Japan; kana6723@yahoo.co.jp (K.Y.); masahiko@v101.vaio.ne.jp (M.S.)

**Keywords:** cytokines, diabetic macular edema, inflammation, anti-VEGF therapy

## Abstract

Diabetic macular edema (DME) is a critical complication of diabetic retinopathy, a condition that arises from the breakdown of the blood–retinal barrier and the consequent increase in vascular permeability. Over the years, attempts have been made to treat DME by various approaches, including laser photocoagulation, steroid triamcinolone acetonide, and vitrectomy. However, treatment was unsatisfactory until research identified vascular endothelial growth factor (VEGF) as a factor in the pathogenesis of DME. Intraocular anti-VEGF agents show good efficacy in DME. Nevertheless, in some patients the condition recurs or becomes resistant to treatment, suggesting that other factors may be involved. Because inflammation and retinal hypoxia are seen in DME, research has examined the potential role of cytokines and other inflammatory mediators. In this review, we provide an overview of this research and describe feedback mechanisms that may represent a target for novel treatments.

## 1. Introduction

Diabetic retinopathy is common in people with type 2 diabetes (T2D) and results from a disruption of the blood–retinal barrier (BRB), which leads to higher vascular permeability and diabetic macular edema (DME). In high-income countries, DME is the most common cause of impaired vision in people with T2D, and in low-income countries it is the most common cause of impaired vision overall [1]. In macular edema, fluid collects in the inner to outer plexiform retinal layer, a change that is hypothesized to result from altered retinal blood flow [2]. Initial treatments for DME included laser photocoagulation, vitrectomy, and steroid triamcinolone acetonide [3,4,5,6]. However, treatment results were unsatisfactory until researchers elucidated the role of the cytokine vascular endothelial growth factor (VEGF) in DME [7]. Successful development of anti-VEGF agents, e.g., aflibercept and ranibizumab, gave physicians a tool to treat DME and improve visual acuity in patients [8,9,10]. Despite these successes, and the clear involvement of VEGF in DME, some patients do not respond to anti-VEGF therapy, or they develop DME again after treatment. These issues have led researchers to hypothesize that cytokines and other inflammatory factors may play a part in DME. This review provides an overview of the pathogenesis of the condition and the purported role of VEGF and other factors and cytokines.

## 2. Pathogenesis

### 2.1. Biochemical Pathways

DME can occur in both proliferative diabetic retinopathy and non-proliferative diabetic retinopathy, suggesting that it is independent of retinopathy progression. One of the proposed mechanisms of DME is that hyperglycemia causes abnormalities of biochemical pathways, i.e., it induces different, overlapping metabolic pathways—the polyol pathway, formation of advanced glycation end-products (AGEs), and activation of protein kinase C (PKC)—and thus initiates the development of a cascade that culminates in the development and progression of diabetic retinopathy [11,12]. For example, the polyol pathway induces oxidative stress, which in turn damages retinal cells and induces diabetic retinopathy. Long-term hyperglycemia leads to the production of AGEs, and the accumulation of AGEs increases the activity of PKC, which in turn causes an upregulation of factors, such as VEGF. Thus, these abnormalities in biochemical pathways can result in an increase of cytokines like VEGF [13]. Indeed, VEGF levels were found to be significantly higher even in diabetic eyes without retinopathy than in non-diabetic controls [14].

In macular edema, more liquid moves from blood vessels into the retinal extracellular space than vice versa [15,16]; this imbalance is assumed to be related to a T2D-related disturbance in the metabolism of Müller cells, which actively transport liquid from the extracellular space into retinal vessels. The resulting dysfunction of interstitial liquid homeostasis and potassium enables the intracellular accumulation of liquid but hinders discharge of the liquid into the vessels. Consequently, cellular swelling occurs, leading to cell rupture and an increase of liquid in the extracellular space, which develops cyst spaces [17]. Fluid accumulates mainly in the extracellular layer of the external plexiform and the inner and outer nuclear retinal layers. The presence of lesions in patients with diabetes leads to a decrease in the number of pericytes, Müller cells, astrocytes, and endothelial cells, as well as an increase in the number of basal membrane capillaries; these changes result in rupture of the BRB and hyperpermeability of retinal vessels [11,18,19,20]. Clinical signs of diabetic retinopathy become apparent only after rupture of the BRB, which is why the development of DME is independent of the progression of diabetic retinopathy. This process results in a loss of endothelial cells and pericytes and an increase of basal membrane capillaries, and together these changes impair retinal blood flow. In summary, the events described above cause retinal hypoxia and disrupt the BRB, leading in turn to further increased production of VEGF or macular edema or both (Figure 1).

### 2.2. VEGF and VEGF Receptors

Increased expression of VEGF is regulated by hypoxia-induced factor1𝛼 (HIF1𝛼). In hypoxic conditions, HIF1𝛼 levels increase, which results in the activation of genes that produce proangiogenic factors (e.g., VEGF) [21,22,23,24]. After exposure to hypoxia, VEGF is expressed by various retinal cells, including Müller cells, as well as lymph nodes, glial cells, retinal pigment epithelium, endothelial cells, and pericytes [25]. Permeability of the vascular endothelium and proliferation of endothelial cells are increased by VEGF [26]. The former results from VEGF-supported changes in the arrangement of cytoplasmic actin filaments and increases in the phosphorylation of proteins, such as zonula occludens-1 and occludin in the tight junctions between cells [27]. Thus, VEGF plays a central role in BRB disruption and macular edema [28,29,30]. In non-human primates, intravitreal injection of VEGF every 3 days for about 2 weeks resulted in a variety of changes to vessels and vascularization that reflected findings in patients with diabetic retinopathy [31]. VEGF was detected in vitreous fluid samples obtained during vitreous surgery to treat DME, so we attempted to treat DME by inhibiting VEGF, and found that DME improved [32].

We previously reported that vitreous VEGF levels were significantly higher in patients with DME than in control patients with nonischemic diseases, such as macular hole [31]. VEGF levels are not necessarily elevated in the eyes of all patients with DME [33], and actually show wide variation among patients. In an earlier study, we evaluated fluorescence in DME that showed that, compared with patients with minimal fluorescence, patients with hyperfluorescence had significantly elevated VEGF levels in the vitreous humor that correlated significantly with the severity of macular edema [31]. On the basis of these results, we hypothesized that diabetes-related retinal hypoxia leads to an increase in VEGF expression, which subsequently damages the BRB and causes macular edema to arise and progress. As described above, VEGF levels increase as a result of hyperglycemia-related changes in biochemical pathways. Therefore, we propose that retinal hypoxia and abnormal biochemical pathways (resulting from hyperglycemia) increase the vitreous levels of VEGF and lead to DME (Figure 1).

VEGF is considered to have a central role in the onset of macular edema, because it strongly increases vascular permeability. Importantly, to mediate its biological effects, VEGF must bind to the VEGF receptor to activate signaling pathways.

Two VEGF receptors—VEGFR-1 and VEGFR-2—are expressed in the retina. Binding of VEGF to either of these receptors activates autophosphorylation [34,35,36], and transphosphorylation induces a signaling cascade. We previously reported that aqueous humor levels of both soluble VEGFR-1 (sVEGRF-1) and sVEGFR-2 were significantly higher in patients with DME than in controls [37]. Soluble VEGFR receptors antagonize the pro-angiogenic and proinflammatory effects of VEGF [38]. However, because levels of soluble VEGF receptors are correlated with levels of transmembrane ones [39], the finding of higher soluble receptor levels suggests that levels of membrane-bound receptors also are higher in patients with DME. Therefore, we hypothesize that these receptors could play an important role in the pathogenesis of DME. VEGFR-1 is mainly expressed by monocytes and macrophages, and VEGFR-1 signaling plays a role in recruitment of leukocytes to sites of inflammation [40]. As has been mentioned before, placental growth factor (PlGF) is a member of the VEGF family [41,42], and like VEGF, also specifically binds to VEGFR-1, stimulating tissue factor production and chemotaxis by monocytes and macrophages [43]. Binding of PlGF to VEGFR-1 increases the production of proinflammatory factors by cultured monocytes via a calcineurin-dependent pathway [44], suggesting that PlGF has a direct influence on the inflammatory response. Thus, the activation of VEGFR-1 promotes inflammation [43,45,46]. On the other hand, VEGFR-2 is exclusively expressed by endothelial cells. Binding of VEGF to VEGFR-2 initiates signaling that not only increases vascular permeability but also upregulates the expression of inflammatory cytokines (such as monocyte chemoattractant protein (MCP)-1 and intercellular adhesion molecule 1 (ICAM-1)) via nuclear factor kappa-light-chain-enhancer of activated B cells (NF-κB) [47,48,49]. We previously suggested that in DME, inflammatory factors may be induced via NF-κB, because we found that the level of sVEGFR-2 was significantly correlated with the levels of several inflammatory factors, including interleukin and MCP-1 [37]. These factors and cytokines induce chemotaxis of leukocytes and promote adhesion of inflammatory cells to the vascular endothelium, resulting in a further increase of vascular permeability. Thus, macular edema is promoted by VEGF binding to its receptors on vascular endothelial cells, monocytes, and macrophages. In addition, clinical and experimental evidence shows that both sVEGFR-1 and sVEGFR-2 may influence vascular permeability during the inflammatory response, because VEGF acts as a chemotactic factor for inflammatory cells via its receptors [50], suggesting that, besides increasing vascular permeability, VEGF may also promote inflammation. Thus, VEGF plays multiple important and complex roles in the pathogenesis of DME [19,51].

### 2.3. Inflammation

Anti-VEGF therapy is ineffective in some patients with DME [10,52]. For example, post hoc exploratory analyses of a study found that, in patients with persistent DME at 24 weeks after ranibizumab treatment, DME was still present in 100% of patients at week 32 and in about 40% at the 3-year visit [53]. In another study, DME recurred in 44 of 68 (64.7%) eyes after the first injection of ranibizumab [32]. These findings suggest that other factors or cytokines may also have a role in the disease (Figure 2).

The potential role of inflammation deserves consideration, because a multicenter clinical trial by the Diabetic Retinopathy Clinical Research Network (DRCR) showed that intravitreal steroid therapy is effective in patients with DME (DRCR.net). Furthermore, the intraocular levels of placental growth factor (PlGF), platelet-derived growth factor (PDGF), interleukin (IL)-6, IL-8, monocyte chemoattractant protein (MCP)-1, intercellular adhesion molecule 1 (ICAM-1), interferon-inducible 10-kDa protein (IP-10), and erythropoietin (EPO) were shown to be higher in patients with DME [7,20,33,37,54,55,56,57]. Studies have shown that EPO may help protect the BRB against disruption [58,59]. The other inflammatory mediators and cytokines, as well as other putative mechanisms of DME, are described in more detail in the next section.

#### Effect of Inflammation on Blood Flow

Inflammation involves the accumulation of leukocytes, e.g., lymphocytes and macrophages, at a lesion, which results in a localized slowing of blood flow. Therefore, the finding that blood flow is slower in eyes with DME than in healthy eyes supports the involvement of inflammation in DME [60,61,62]. In a previous study with scanning laser ophthalmoscopy and fluorescein angiography, we were able to assess blood flow velocity by measuring the speed of particles, proposed to be leukocytes, in the capillaries [63,64]. In a further study, we showed that retinal blood flow velocity is decreased in patients with T2D and DME compared with both patients with T2D and no DME and healthy controls, and we found a correlation between the extent of the slowing of blood flow and the severity of DME [65]. We interpreted our results as indicating that DME involves an inflammatory state and proposed that the inflammation was a response to damage to the vascular endothelium. Specifically, we suggested that in DME, expression of inflammatory factors increases in response to endothelial cell damage, and the resulting leukocyte rolling and adhesion decreases blood flow velocity. Another study of ours provided evidence to support this hypothesis, in that levels of inflammatory factors, including MCP-1 and IL-8, correlated negatively with blood flow velocity speed [66].

## 3. Soluble Mediators Involved in DME

### 3.1. Growth Factors

#### 3.1.1. PlGF

PlGF, a 25-kd dimeric protein, is a member of the VEGF family and highly homologous with VEGF [41,42]. As a specific ligand for VEGFR-1, PlGF potently promotes angiogenesis, and also induces the growth and migration of endothelial cells [67]. Similar to VEGF, PlGF binds exclusively to VEGFR-1 [68]; however, it shows higher affinity for this receptor than VEGF does [69]. PlGF also modulates the inflammatory process by stimulating tissue factor production and chemotaxis in monocytes and macrophages [43]. Previously, we showed a significant correlation between the PlGF level in aqueous humor and DME severity [37]. This finding is supported by the finding that intravitreously injected PlGF-1 causes the external retinal barrier to rupture and also leads to retinal edema [70]. Taken together, these findings indicate that PlGF may be involved in macular edema in DME.

#### 3.1.2. PDGF

PDGF is a member of the PDGF/VEGF family, and is involved in regulating the migration of various mesenchymal cells, including fibroblasts, glial cells, and smooth muscle cells [71,72]. PDGF is a dimeric protein that exists in various isoforms, and the isoform PDGF-AA was reported to induce gap bond formation [71]. PDGF is secreted by endothelial cells and is responsible for the function of pericytes [73]. PDGF was also reported to be a hypoxia-regulated gene product that, along with VEGF, contributes to ocular neovascularization [74]. These findings suggest that the PDGF/VEGF family may act synergistically in patients with DME. Previously, we showed a significant correlation between PDGF, VEGF, and PlGF in patients with DME [37]. We interpreted this finding as indicating that in DME, levels of these three growth factors in the aqueous humor are closely related. Support for this hypothesis was provided by the finding that, together with VEGF and PlGF, PDGF probably plays a role in regulating the survival of pericytes and endothelial cells [75]. The finding that PDGF-AA in aqueous humor correlates significantly with DME severity also suggests that PDGF may affect macular edema in DME [37].

### 3.2. Cytokines and Chemokines

#### 3.2.1. IL-6

The cytokine IL-6 has many functions. For example, it rearranges actin filaments, which creates intercellular gap junctions and consequently increases the permeability of the endothelial cells lining blood vessels [76]. In vitro, hypoxic conditions gradually increase the expression of IL-6 mRNA by endothelial cells [77,78,79]. One study showed an association between IL-6 levels and risk for DME, so IL-6 may potentially be a predictor or therapeutic target in this disease [80].

#### 3.2.2. IL-8

Interleukin (IL)-8, a potent chemoattractant, promotes an inflammatory response by activating both neutrophils and T cells. In vitro, both hypoxia and oxidative stress cause vascular endothelial cells to produce IL-8 [81,82]. Furthermore, IL-8 has effects on intracellular tight junctions, downregulating them to increase vascular permeability [83,84]. A study in patients with DME found lower mean IL-8 levels in those who responded to treatment than in those who did not; a multivariate logistic regression analysis of various factors found that only IL-8 was related to treatment response, so IL-8 levels in aqueous humor may reflect response to treatment with anti-VEGF agents in DME [85].

#### 3.2.3. MCP-1

Expression of the chemokine MCP-1, which increases monocyte chemotaxis, is elevated in hypoxic conditions in the retina, as well as in the presence of arteriosclerosis and oxidative stress [86,87,88]. Similar to Il-8, MCP-1 opens tight junctions by promoting phosphorylation of junction proteins, e.g., zonula occludens-1 and occludin [89,90]. A clinical study found that MCP-1 levels were lower in patients who responded to treatment than in those who did not, and the authors suggested that baseline MCP-1 may therefore be a predictor of response to anti-VEGF treatment [91].

#### 3.2.4. IP-10

IP-10 activates the cell-mediated immune response in general, and in particular the T-helper type 1 immune response. It is released by macrophages, endothelial cells, and fibroblasts, and acts as a chemoattractant that attracts a range of cells, including macrophages and T cells. IP-10 inhibits proliferation, induces the apoptosis of endothelial cells, and counteracts the VEGF-related increase in their motility [92,93]. The role of IP-10 in disruption of the BRB is unclear, but previously we reported that IP-10 levels correlated with levels of MCP-1, IL-6, and IL-8, all of which are involved in inflammation [37]. Consequently, we assumed that IP-10 may also be involved in the inflammatory process in DME. In the same study, IP-10 in aqueous humor showed a significant correlation with PlGF and DME severity [37]. On the basis of this study, we hypothesized that IP-10 may be involved in inflammation by increasing leukocyte rolling and adhesion to vessel walls. Therefore, promotion of inflammation by IP-10, as well as PlGF and PDGF-AA, may play a role in macular edema in DME.

### 3.3. Other Mediators

#### ICAM-1

ICAM-1, an adhesion molecule, is a member of the immunoglobulin superfamily and is a ligand for the lymphocyte function-associated antigen 1 receptor. Its expression by various other cells in the retina and choroid (as well as leukocytes) has been demonstrated in vivo and in vitro [94]. In addition, the expression of ICAM-1 mRNA and protein is upregulated by retinal hypoxia [95,96]. As a consequence, ICAM-1 upregulation promotes leukostasis by increasing the rolling of leukocytes, and their adhesion to vessel walls; hypoxia; endothelial cell damage; and breakdown of the BRB—key events in DME pathogenesis [97,98,99,100]. Furthermore, in the diabetic rat model, blockade of the ICAM-1 and lymphocyte function-associated antigen 1 axis prevented retinal leukostasis and promoted BRB integrity [98,101]. A study in patients with DME found that higher baseline ICAM-1 levels in aqueous humor were associated with good treatment response to ranibizumab [102].

Inflammation is likely involved in the pathogenesis of DME, because intraocular levels of the inflammatory factors and cytokines described above are increased in patients with DME. The aqueous flare is an index of ocular inflammation that can be measured with a laser flare meter, and aqueous flare values have been reported to be significantly higher in patients with DME than in control patients [103], providing evidence of active inflammation in DME. In addition, the aqueous flare value shows a significant positive correlation with the intraocular levels of inflammatory cytokines [103], so a higher flare value is associated with higher levels of inflammatory cytokines. Moreover, not only the VEGF level, but also the levels of the above-mentioned factors and cytokines are significantly higher in people with hyperfluorescent DME than in those with minimally fluorescent DME [33]. These findings are supported by the results of basic studies [76,83,84,89,90].

Accordingly, DME appears to cause both retinal hypoxia and inflammation through hyperglycemia and the resulting abnormalities of biochemical pathways, as detailed above. As a result, expression of VEGF and inflammatory cytokines increases, resulting in disruption of the BRB and development and progression of macular edema (Figure 3).

## 4. Hypothesis of the Mechanism of DME Proposed by the Authors

As described above, hyperglycemia can lead to oxidative stress, formation of AGEs, and activation of PKC. The resulting abnormalities of biochemical pathways could induce expression of VEGF, a loss of endothelial cells and pericytes, and an increase in basal membrane capillaries, resulting in inflammation. The loss of endothelial cells and pericytes and increase of basal membrane capillaries result in impaired retinal blood flow. Furthermore, inflammation involves the activation of various inflammatory factors and cytokines. These events cause retinal hypoxia and disruption of the BRB, leading in turn to further increased production of VEGF, which promotes macular edema. VEGF and PlGF both activate VEGFR-1, which increases levels of inflammatory cytokines and supports leukocyte recruitment. VEGF also activates VEGFR-2, which increases vascular permeability and activates NF-κB, which in turn increases the expression of inflammatory cytokines, such as MCP-1 and ICAM-1. These processes result in leukocyte chemotaxis and inflammatory cell adhesion at the vascular endothelium, decreasing local blood flow velocity, further increasing retinal hypoxia, and creating a positive feedback loop (positive feedback loop 1) [104]. The leukocyte chemotaxis and inflammatory cell adhesion themselves also increase inflammation, creating a second positive feedback loop (positive feedback loop 2), whereby higher levels of inflammatory cytokines, e.g., MCP-1 and ICAM-1, cause leukocyte abnormalities and reduce retinal blood flow velocity (Figure 4).

As a result of positive feedback loops 1 and 2, the inflammatory state increases over time and VEGF levels increase, causing chronic, refractory DME that is harder to treat. Consequently, anti-VEGF therapy should be initiated as early as possible to prevent these feedback loops from becoming established.

## 5. Therapy

Anti-VEGF therapy is more effective than laser alone for the treatment of DME [105,106,107,108,109,110], providing further support for the hypothesis that the earlier treatment begins, the better the improvement in visual acuity. According to our hypothesis described above, early anti-VEGF therapy should break the two positive feedback loops, i.e., the further exacerbation of retinal hypoxia by reduction of blood flow velocity and the enhancement of inflammation by increased leukocyte chemotaxis and adhesion.

Longer-term studies of anti-VEGF therapy have shown that visual acuity gained by year 1 was maintained at years 3 or 5 [111,112]. These studies also reported a decrease in the frequency of injections during the five-year study period. Together, these findings suggest that early anti-VEGF therapy blocks the positive feedback loops. Thus, inhibition of VEGF by anti-VEGF agents results in decreased leukocyte rolling and adhesion (leukostasis) and prevents the positive feedback loops from being established. Anti-VEGF therapy inhibits the positive feedback loops because, in addition to reducing levels of VEGF itself, it decreases the levels of downstream inflammatory cytokines. Campochiaro et al. [104] reported that the percentage of patients with no retinal nonperfusion (RNP) decreased in the sham group between baseline and month 24, and that initiation of ranibizumab treatment in the sham group at month 24 was followed by a reduction in the percentage of patients with an increase in RNP from baseline at months 30 and 36, suggesting that aggressive VEGF blockade prevents the progression of RNP and promotes reperfusion by controlling positive feedback loop 1. In addition, the aqueous flare value showed a significant decrease 1 month after anti-VEGF therapy [113], suggesting that this therapy inhibits inflammation by controlling positive feedback loop 2. Accordingly, anti-VEGF therapy seems to improve retinal hypoxia and inflammation in patients with DME by inhibiting both VEGF and inflammatory cytokines, thus suppressing leukocyte chemotaxis and adhesion in the retinal vessels.

The DRCR Protocol T study performed a head-to-head comparison of bevacizumab, ranibizumab, and aflibercept, and confirmed that all three drugs were effective in patients with DME [10]. The improvement in visual acuity at 1 year was significantly greater in patients in the aflibercept group, who had an initial visual acuity of 20/50 or worse than in their counterparts in the bevacizumab and ranibizumab groups. On the basis of our hypothesis, we propose that aflibercept might be useful in severe cases, because it also binds to PlGF, resulting in the suppression of various inflammatory factors.

If anti-VEGF therapy is delayed, the positive feedback loops advance, and not only VEGF but also inflammatory factors are expressed, resulting in refractory macular edema. Therefore, as mentioned before, anti-VEGF therapy should be started as soon as possible. If therapy is delayed, steroid therapy may be required in patients with high levels of inflammatory factors, because steroids can decrease the expression of such factors [114,115]. Furthermore, steroids restore metabolic functions, helping restore potassium and promote aqueous drainage from the retina into the vessels [17]. In fact, in pseudophakic eyes, triamcinolone (plus prompt laser) was more effective than sham (plus prompt laser) and as effective as ranibizumab (plus prompt laser). In pseudophakic eyes, steroid injections and anti-VEGF injections may be similarly effective for improving visual acuity and decreasing retinal thickening [116]. As mentioned above, cytokines are expressed before the onset of DME. Therefore, DME is already considered to have a high specific gravity for inflammation and to be a consequence of the chronic inflammatory state caused by diabetes. In eyes with persistent DME, the mean change in central subfield thickness was found to be better in the combination group (ranibizumab plus intravitreous dexamethasone implant) than in the ranibizumab group (intravitreous ranibizumab alone) [52], suggesting that steroids can improve macular edema because they suppress various inflammatory. In addition, Shimura et al. [117] reported that eyes treated by sub-Tenon’s capsule triamcinolone acetonide injection (STTA) showed significantly more regression of macular edema and improvement of visual acuity than did the controls, and that the mean number of bevacizumab injections required in the STTA-treated eyes was significantly lower than in the control eyes. Therefore, combined treatment with steroids may be useful in cases with high levels of inflammatory factors.

Interestingly, treatment with ranibizumab and aflibercept was recently reported to decrease the severity of diabetic retinopathy, suggesting that anti-VEGF agents may help to prevent the progression of diabetic retinopathy [108,118,119,120]. Furthermore, the DRCR Protocol I showed that in the five-year completers, the median number of injections in year 5 was 0 in both the group with ranibizumab and prompt laser and the group with ranibizumab and deferred laser [112]. If we relate these findings to our hypothesis, we suggest that continuing anti-VEGF therapy may be able to stop the positive feedback loops and improve DME.

The core RIDE and RISE studies found a trend towards an increase in the development of new, proliferative diabetic retinopathy events over 4 years, despite performing anti-VEGF therapy [121]. Interestingly, Couturier et al. [122] recently reported that after three anti-VEGF injections in eyes with diabetic retinopathy, neither ultra-wide-field fluorescein angiography nor swept-source wide-field OCT angiography detected reperfusion of vessels or the capillary network in nonperfusion areas. Furthermore, the group observed no reperfusion, even when a reduction in dark areas was visible on fluorescein angiography, although they reported that RNP improved after anti-VEGF therapy [104]. These findings and our results suggest that anti-VEGF agents do not improve retinal blood flow. This hypothesis is supported by the finding that both retinal and choroidal blood flow significantly decreased after anti-VEGF therapy [66]. Therefore, retinal photocoagulation might need to be performed for retinal ischemia. In fact, central macular thickness was found to increase gradually in patients who did not receive targeted retinal photocoagulation, but not in those who did, indicating that targeted retinal photocoagulation may prevent the recurrence of macular edema [123].

## 6. Potential Novel Drug Targets Associated with Mediators

### 6.1. VEGF Designed Ankyrin Repeat Protein

Designed ankyrin repeat proteins (DARPins) are genetically engineered and act like antigens, in that they bind to target proteins with high affinity and in a highly specific way. A VEGF DARPin has been tested for treatment because it is highly potent and has a long half-life, meaning that injection intervals may be longer than those with standard treatment. A phase I/II study in DME found that the VEGF DARPin abicipar pegol had similar effects as monthly ranibizumab injections in improving vision and reducing central retinal thickness [124]. A phase III study on DME is currently being planned [125].

### 6.2. Interleukin Inhibitors

Among the interleukins, IL-6 is assumed to be involved in DME, because vitreous humor levels were repeatedly found to be increased. Therefore, research is being performed on the efficacy of anti-IL-6 treatment in DME. One clinical study evaluated the IL-6 humanized monoclonal antibody EBI-031 (Eleven Biotherapeutics, Roche, Cambridge, Mass, United States), which is hypothesized to bind to IL-6 and prevent IL-6 intraocular signaling activity. Another study, Ranibizumab for Edema of the mAcula in Diabetes: Protocol 4 with Tocilizumab (READ-4), is comparing intravenous administration of the humanized monoclonal IL-6 antibody tocilizumab with ranibizumab alone and in combination in patients with DME. Tocilizumab is used as an immunosuppressant treatment for non-infectious uveitis [126].

### 6.3. Inhibitors of Adhesion Molecules

Vascular adhesion protein (VAP)-1 is an endothelial adhesion molecule similar to ICAM-1. Endothelial cells express VAP-1, and it is involved in inflammation by supporting the transmigration of leukocytes. An oral VAP-1 inhibitor was tested in a rat model of diabetes and reduced increased vascular permeability in the retina [127]. In patients with DME, the oral VAP-1 inhibitor ASP8232 (Astellas, Northbrook, IL, USA) was compared with ranibizumab in a three-arm phase II study (oral ASP8232 + sham injection, ASP8232 + ranibizumab injection, oral placebo + ranibizumab injections) [128]. However, although ASP8232 almost fully inhibited activity of VAP-1 in plasma, no effects were seen on DME parameters, and the combination treatment showed no advantage over monotherapy with ranibizumab [128]. Despite these disappointing findings, research is still being performed on intravitreal VAP-1 inhibitors, in the hope that they may nevertheless be useful in managing DME.

### 6.4. Inhibitors of Multiple Growth Factors

The antiviral and anti-angiogenic drug squalamine inhibits several growth factors involved in angiogenesis, including VEGF and PDGF [129]. Not all VEGF-mediated inflammatory pathways are blocked by squalamine, but it does affect the mitogen-activated protein kinase, p38 inflammatory, and vascular endothelial–cadherin r signaling pathways, among others. A study in patients with macular edema due to retinal vein occlusion showed better visual improvements with a combination of topical squalamine (0.2%; Ohr Pharmaceutical, New York, NY, USA) and early, as-needed injection of ranibizumab than with ranibizumab alone [130]. Currently, a phase II study in patients with DME is evaluating the efficacy of squalamine eye drops.

## 7. Conclusions

This paper reviews the involvement of cytokines in the pathogenesis of DME. Research shows that in DME hyperglycemia causes abnormalities of biochemical pathways, that VEGF and various inflammatory cytokines promote the development of inflammation and retinal hypoxia, and that the underlying mechanism is complex. The positive feedback loops form over time, and consequently, the expression of inflammatory cytokines increases and inflammation worsens. The disease becomes resistant to anti-VEGF therapy, and thus more difficult to treat. The cytokine hypothesis for the pathogenesis of DME seems to be plausible considering the available data, and could also be useful for developing new therapeutic strategies.

## Figures and Tables

**Figure 1 ijms-22-03427-f001:**
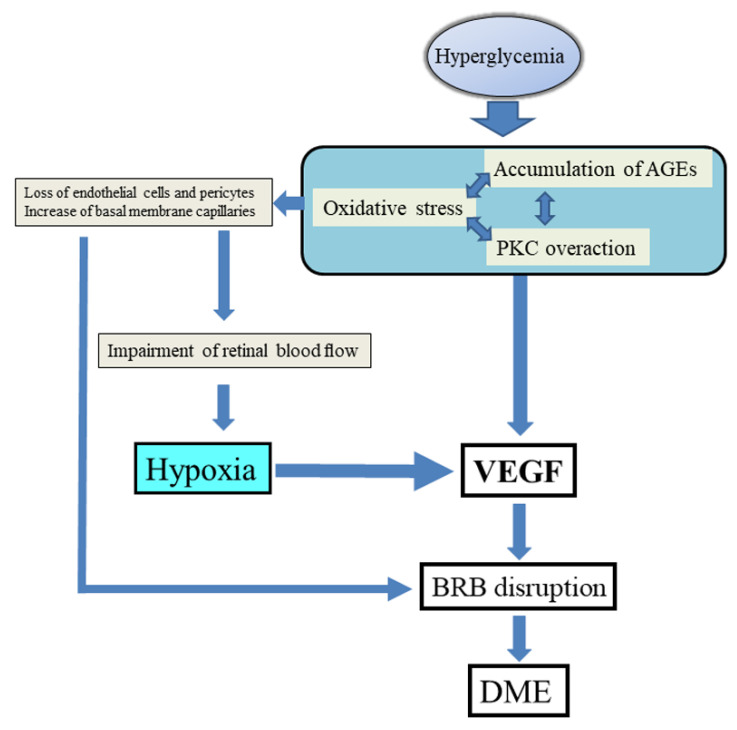
Hypothesized pathogenesis of diabetic macular edema. Hyperglycemia induces oxidative stress and, if maintained over time, produces advanced glycation end-products (AGEs). The accumulation of any AGEs increases the activity of protein kinase C (PKC), which in turn causes upregulation of various factors, including vascular endothelial growth factor (VEGF). A loss of endothelial cells and pericytes and increase of basal membrane capillaries result in impairment of retinal blood flow. These events cause retinal hypoxia and increase vascular permeability, leading in turn to further increased production of VEGF or macular edema or both. Abbreviations: BRB, blood-retina barrier; DME, diabetic macular edema.

**Figure 2 ijms-22-03427-f002:**
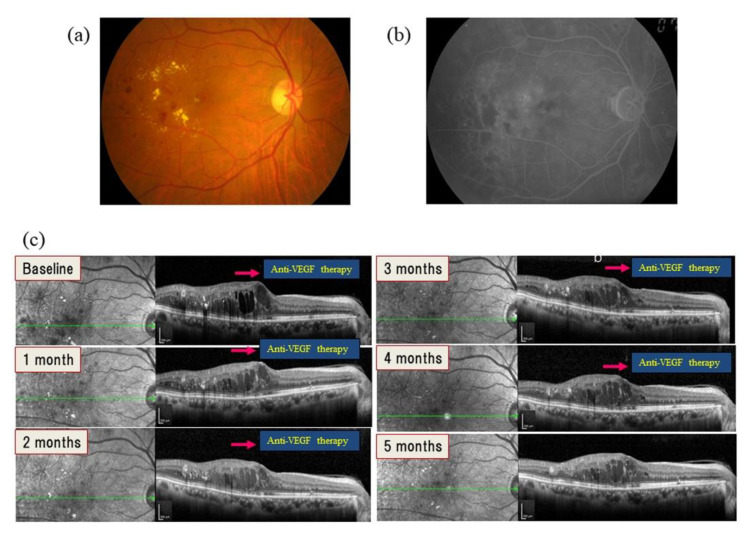
Images from a representative patient with diabetic macular edema (DME; male, 64 years old) in whom anti-vascular endothelial growth factor therapy was ineffective for diabetic macular edema. (**a**) Color fundus photograph of diabetic macular edema (DME) at baseline. (**b**) Fluorescein angiogram showing DME at baseline. (**c**) Optical coherence tomography images of DME from baseline to 5 months. DME did not improve despite five consecutive monthly doses of anti-vascular endothelial growth factor (anti-VEGF) therapy. Aqueous humor was collected at baseline and at the fifth injection for assessment of VEGF and other cytokines. Inflammatory factors were found to be elevated even though VEGF levels decreased (VEGF, 51.4 pg/mL → 0 pg/mL; monocyte chemoattractant protein (MCP)-1, 1601 pg/mL → 1915 pg/mL; intercellular adhesion molecule 1 (ICAM-1), 18.58 pg/mL → 22.72 pg/mL; interleukin (IL)-6, 8.86 pg/mL→ 30.23 pg/mL; IL-8, 9.80 pg/mL → 11.8 pg/mL). This finding suggests that other factors or cytokines besides VEGF may also have a role in the development of DME.

**Figure 3 ijms-22-03427-f003:**
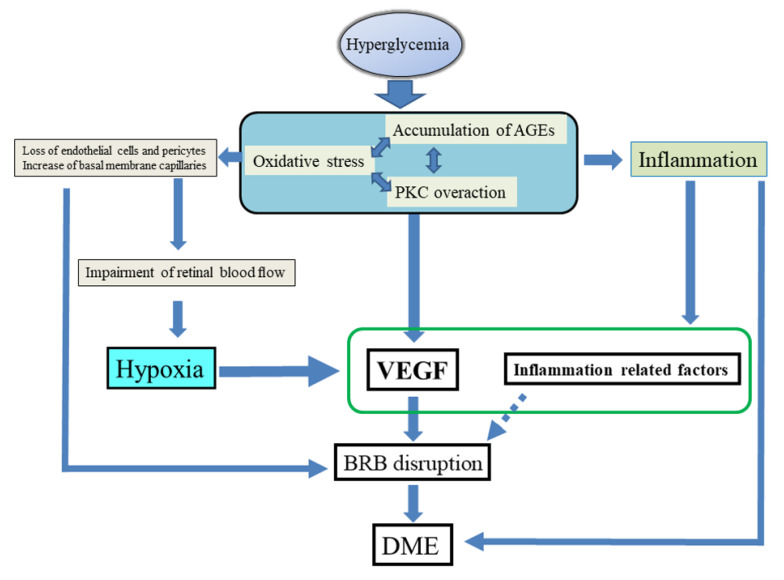
Hypothesized pathogenesis of diabetic macular edema. In diabetic macular edema (DME), hyperglycemia causes abnormalities of biochemical pathways, leading to both retinal hypoxia and inflammation. As a result, expression of vascular endothelial growth factor (VEGF) and inflammatory cytokines are increased, resulting in disruption of the blood–retina barrier (BRB) and development and progression of macular edema. Abbreviations: AGEs, advanced glycation end-products; PKC, protein kinase C.

**Figure 4 ijms-22-03427-f004:**
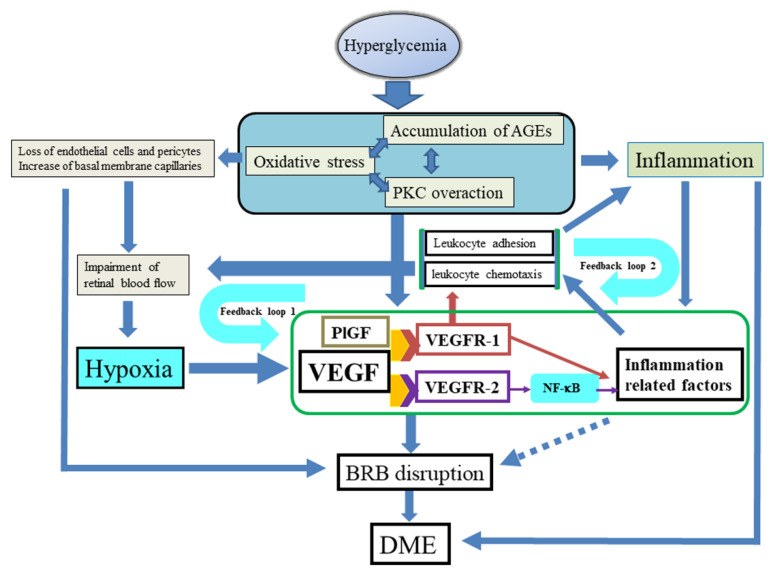
Hypothesized pathogenesis of diabetic macular edema. Hyperglycemia can lead to oxidative stress, formation of advanced glycation end-products (AGEs), and activation of protein kinase C (PKC). These abnormalities in biochemical pathways could induce the expression of vascular endothelial growth factor (VEGF), loss of endothelial cells or pericytes, an increase of basal membrane capillaries, and inflammation. The loss of endothelial cells and pericytes and increase in basal membrane capillaries may result in impairment of retinal blood flow. Furthermore, inflammation involves activation of various inflammatory factors and cytokines. These events cause retinal hypoxia and disrupt the blood–retina barrier (BRB), leading in turn to further increased production of VEGF and the development and progression of macular edema. Moreover, activation of VEGFR-1 by both VEGF and placental growth factor (PlGF) plays a role in the recruitment of leukocytes, and also upregulates the expression of inflammatory cytokines. Activation of VEGFR-2 by VEGF increases vascular permeability and enhances the expression of inflammatory cytokines, such as monocyte chemoattractant protein 1 (MCP-1) and intercellular adhesion molecule 1 (ICAM-1) via NF-κB, leading to chemotaxis and adhesion of leukocytes to the vascular endothelium, along with a decrease of blood flow velocity. Reduction of blood flow velocity creates positive feedback loop 1, which further exacerbates retinal hypoxia. Increased leukocyte chemotaxis and adhesion also enhances inflammation, creating positive feedback loop 2.

## Data Availability

Data sharing not applicable.

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
