# Peer review of "Involvement of Cytokines in the Pathogenesis of Diabetic Macular Edema"

_ijms, 2021, doi:10.3390/ijms22073427_

Round 1

Reviewer 1 Report

I accept the revision and recommend for publication.

Reviewer 2 Report

No more comments.

This manuscript is a resubmission of an earlier submission. The following is a list of the peer review reports and author responses from that submission.

Round 1

Reviewer 1 Report

In this review, Noma et al. provide an overview on diabetic macular edema (DME), a complication of diabetic retinopathy (DR). The authors describe the role of inflammation and soluble mediators in DME pathogenesis and the therapeutic treatments available. Finally, they propose an hypothesis of mechanism of DME pathogenesis, taking into account all the alterations that occur in the retina of DR patients.

This review is certainly interesting. However, I found the manuscript difficult to follow, and I strongly believe that the logic flow should be modified to make the review clearer.

Major concerns:

1) As stated above, the author should modify the flow of the manuscript in order to make it more logic and consistent with the title of their review. Here below a suggestion of a possible alternative outline (that is, as I said, just a suggestion).

1 introduction

2 pathogenesis

2.1 biochemical pathways

2.2 VEGF (and VEGFRs, of course)

2.3 inflammation

2.3.1 effect of inflammation on blood flow

3 soluble mediators involved in DME

3.1 growth factors [other than VEGF, i.e. PlGF, PDGF, EPO (why EPO has not been included in growth factor section?)]

3.2 cytokines and chemokines (IL6, IL8, IP-10, MCP-1)

3.3. other mediators (ICAM-1)

4 therapy

5 proposed hypothesis of mechanism by the authors

6 conclusions

2) Is not clear to me why the authors introduced the role of VEGF in DME describing their attempt to treat DME with anti-VEGF. I would start the chapter from line 77.

3) How many patients do not respond to anti-VEGF therapy? And how many experience pathology relapse? Please add some details.

4) May you explain better which is the role of IP-10 and EPO in BRB disruption?

5) As for the high levels of VEGFR1 and VEGFR2 found in aqueous humor, do the authors refer to the soluble forms of the receptors? If yes, since the soluble receptors act as decoy receptors, i.e. antagonizing the pro-angiogenic/proinflammatory effect of VEGF, may the authors explain which should be their role in favoring edema formation?

Minor points:

6) please replace “multiplication” with “proliferation” at line 82 and elsewhere in the manuscript

7) please, pay attention to the correct nomenclature of NF-κB (e.g. B, not β) and make it consistent throughout the paper – lines 235 – 235 – 275 – 301 etc

8) line 218: please rephrase

Author Response

Response to Reviewer 1 Comments

In this review, Noma et al. provide an overview on diabetic macular edema (DME), a complication of diabetic retinopathy (DR). The authors describe the role of inflammation and soluble mediators in DME pathogenesis and the therapeutic treatments available. Finally, they propose an hypothesis of mechanism of DME pathogenesis, taking into account all the alterations that occur in the retina of DR patients.

This review is certainly interesting. However, I found the manuscript difficult to follow, and I strongly believe that the logic flow should be modified to make the review clearer.

Major concerns:

Point 1: As stated above, the author should modify the flow of the manuscript in order to make it more logic and consistent with the title of their review. Here below a suggestion of a possible alternative outline (that is, as I said, just a suggestion).

1 introduction

2 pathogenesis

2.1 biochemical pathways

2.2 VEGF (and VEGFRs, of course)

2.3 inflammation

2.3.1 effect of inflammation on blood flow

3 soluble mediators involved in DME

3.1 growth factors [other than VEGF, i.e. PlGF, PDGF, EPO (why EPO has not been included in growth factor section?)]

3.2 cytokines and chemokines (IL6, IL8, IP-10, MCP-1)

3.3. other mediators (ICAM-1)

4 therapy

5 proposed hypothesis of mechanism by the authors

6 conclusions

Response 1: Thank you so much for this helpful suggestion. As far as possible, we have rearranged the manuscript according to your suggestions.

Point 2: Is not clear to me why the authors introduced the role of VEGF in DME describing their attempt to treat DME with anti-VEGF. I would start the chapter from line 77.

Response 2: Thank you for this comment. We agree with your suggestion to start this chapter at line 77. However, rather than deleting the text in lines 72 through 76, which highlights our earlier work, we decided to incorporate it later in the chapter (page 2, lines 77-93). We hope that you agree with this solution.

Point 3: How many patients do not respond to anti-VEGF therapy? And how many experience pathology relapse? Please add some details.

Response 3: Thank you for this suggestion. We have added the following information to the manuscript (page 4, lines 151-154):

" For example, post hoc exploratory analyses of a study found that, in patients with persistent DME at 24 weeks after ranibizumab treatment, DME was still present in 100% of patients at week 32 and in about 40% at the 3-year visit (Bressler, JAMA Ophthal 2016).

In another study, DME recurred in 44 of 68 (64.7%) eyes after the first injection of ranibizumab (Shimura, BJO 2017)."

Point 4: May you explain better which is the role of IP-10 and EPO in BRB disruption?

Response 4: Thank you for this valuable suggestion. Unfortunately, we were unable to find any publications on the role of IP-10 in BRB disruption. However, on the basis of our research we hypothesize that IP-10 may disrupt the BRB by engaging in increased leukocyte rolling and adhesion to vessel walls. We have added this hypothesis to the subsection on IP-10 (page 7, lines 265-273).

Regarding EPO, our additional search of the literature found that it actually helps to protect the BRB. Therefore, we deleted subsection 3.6 and changed the final sentence at the end of the introductory paragraph for section 2.3 to the following: "Studies showed that EPO may help protect the BRB against disruption. The other inflammatory mediators and cytokines, as well as other putative mechanisms of DME, are described in more detail next section" (page 5, lines 183-185).

Point 5: As for the high levels of VEGFR1 and VEGFR2 found in aqueous humor, do the authors refer to the soluble forms of the receptors? If yes, since the soluble receptors act as decoy receptors, i.e. antagonizing the pro-angiogenic/proinflammatory effect of VEGF, may the authors explain which should be their role in favoring edema formation?

Response 5: Yes, we are referring to the soluble forms of the receptors. However, in our previous work we showed that levels of soluble VEGF receptors are correlated with levels of  transmembrane ones. Therefore, we propose that high levels of soluble receptors mirror high levels of membrane-bound receptors and, consequently, do not negate the pro-angiogenic/proinflammatory effects of VEGF. We have added this hypothesis to the revised manuscript (page 3, lines 121-125).

Minor points:

Point 6: please replace “multiplication” with “proliferation” at line 82 and elsewhere in the manuscript.

Response 6: Thank you for this comment. As suggested, we have replaced “multiplication” with “proliferation” (page 2, line 82; page 6, line 264).

Point 7: please, pay attention to the correct nomenclature of NF-κB (e.g. B, not β) and make it consistent throughout the paper – lines 235 – 235 – 275 – 301 etc

Response 7: Thank you for this comment. As suggested, we have ensured that the nomenclature of NF-κB is correct (e.g. B, not β) throughout the paper (page 3, lines 137 and 138; page 10, line 389; page 11, line 415).

Point 8: line 218: please rephrase

Response 8: Thank you for this comment. As suggested, we have rephrased the sentence, as follows: "Two VEGF receptors—VEGFR-1 and VEGFR-2—are expressed in the retina" (page 3, line 117).

Reviewer 2 Report

in line 77-79, author said: "Increased expression of VEGF is regulated by hypoxia-induced factor1? (HIF1?). In  hypoxic conditions, HIF1? levels decrease, which results in the activation of genes that  produce proangiogenic factors". 

Question: In hypoxic condiiton, HIF1a will be upregulated, I am not sure why it is reduced in this case, please double check or explain.

This is a review paper regarding the involvement of cytokines in Diabetic macular ademia(DMA), and potential  novel target for drugs. Author should include more recent research reports for cytokines involvement in DMA, I did not see much recent works cited. Author spent significant amount  of time to describe mechanism of DMA, much less portion of this review is associated with cytokines. Author should consider to cite more clinical papers about cytokines role in DMA. If author wants to discuss the potencial novel drug targets associated with cytokines, there should be additional section(or table) for current drug targets, whis is missing in this manuscript.

Author Response

Response to Reviewer 2 Comments

in line 77-79, author said: "Increased expression of VEGF is regulated by hypoxia-induced factor1? (HIF1?). In  hypoxic conditions, HIF1? levels decrease, which results in the activation of genes that  produce proangiogenic factors". 

Question:

Point 1: In hypoxic condiiton, HIF1a will be upregulated, I am not sure why it is reduced in this case, please double check or explain.

Response 1: Thank you for noticing this error, which we have corrected (page 2, line 78).

Point 2: This is a review paper regarding the involvement of cytokines in Diabetic macular ademia (DMA), and potential novel target for drugs. Author should include more recent research reports for cytokines involvement in DMA, I did not see much recent works cited. Author spent significant amount of time to describe mechanism of DMA, much less portion of this review is associated with cytokines. Author should consider to cite more clinical papers about cytokines role in DMA. If author wants to discuss the potencial novel drug targets associated with cytokines, there should be additional section (or table) for current drug targets, which is missing in this manuscript.

Response 2: Thank you for these useful remarks. We have added some more recent reports on the involvement of cytokines, as well as some clinical studies (page 6, lines 212-216, 225-233, and 239-241; page 7, lines 247-251, 257-259, 265-273, and 285-287). Furthermore, we have added some potential novel drug targets associated with mediators (page 12, line 504, to page 13, line 548).
